# From graph topology to ODE models for gene regulatory networks

Xiaohan Kang[1]*, Bruce Hajek[1], Yoshie Hanzawa[2]

**1** Department of Electrical and Computer Engineering, and Coordinated Science Laboratory, University of Illinois at Urbana–Champaign, Urbana, Illinois, United States of America, **2** Department of Biology, California State University, Northridge, Northridge, California, United States of America

* xiaohan.kang1@gmail.com

**Data Availability Statement:** The computer simulation source code is available at https://github.com/Veggente/graph-ode.

**Funding:** This work was supported by the Plant Genome Research Program from the National Science Foundation (NSF-IOS-PGRP-1823145) to

## Abstract

A gene regulatory network can be described at a high level by a directed graph with signed edges, and at a more detailed level by a system of ordinary differential equations (ODEs). The former qualitatively models the causal regulatory interactions between ordered pairs of genes, while the latter quantitatively models the time-varying concentrations of mRNA and proteins. This paper clarifies the connection between the two types of models. We propose a property, called the constant sign property, for a general class of ODE models. The constant sign property characterizes the set of conditions (system parameters, external signals, or internal states) under which an ODE model is consistent with a signed, directed graph. If the constant sign property for an ODE model holds globally for all conditions, then the ODE model has a single signed, directed graph. If the constant sign property for an ODE model only holds locally, which may be more typical, then the ODE model corresponds to different graphs under different sets of conditions. In addition, two versions of constant sign property are given and a relationship between them is proved. As an example, the ODE models that capture the effect of *cis*-regulatory elements involving protein complex binding, based on the model in the GeneNetWeaver source code, are described in detail and shown to satisfy the global constant sign property with a unique consistent gene regulatory graph. Even a single gene regulatory graph is shown to have many ODE models of GeneNetWeaver type consistent with it due to combinatorial complexity and continuous parameters. Finally the question of how closely data generated by one ODE model can be fit by another ODE model is explored. It is observed that the fit is better if the two models come from the same graph.

## Introduction

A gene regulatory network is a collection of molecular classes such that each molecular class interacts with a small number of other molecular classes, creating a sparse graph structure [1]. A goal of systems biology is to understand gene regulatory networks and infer them from data [2, 3]. A directed graph with vertices representing genes and signed edges representing gene-to-gene interactions, also known as a circuit model [4] or a logical model [5], is a model with a high level of abstraction (see S1 Appendix). The vertices of such graph models often only

B.H. and Y.H., and by the Communication and Information Foundations program from the National Science Foundation (NSF-CCF-CIF-1900636) to B.H.

**Competing interests:** The authors have declared that no competing interests exist.

consist of the genes but not the properties of the derived proteins because the latter information is usually not available. An ordinary differential equation (ODE) model is far more detailed than a graph model: they quantitatively describe the dynamics of the time-varying mRNA and protein concentrations of the genes, and can be used to capture complex effects, including protein–protein interaction, post-translational modification, environmental signals, diffusion of proteins in different parts of the cell, and various time constants. As a result, ascribing a directed graph to a biologically plausible gene regulatory network can miss important biological details and dynamics because of the abstraction. However, it is significantly more challenging to ascribe a particular ODE model to a gene regulatory network than to ascribe a directed graph because an ODE model requires much finer classification with possibly orders of magnitude more amount of data. As one example, the work [6] is notable for successful identification of an ODE model that captures the gene regulatory network underlying the dynamics of the circadian clock. The ODE model in [6] is based on a number of previous empirical and modeling studies, and it is shown that parameters for the model can be selected to give a good match to the data. In general, however, without such prior knowledge, the relation between the graph models and the ODE models is unclear. The purpose of this paper is to explore the connections between the two types of models.

We propose a property of the ODE models, called the constant sign property (CSP), such that an ODE model corresponds to a single graph model under a set of conditions if and only if the ODE model satisfies CSP under that set of conditions. An ODE model is said to satisfy global constant sign property (GCSP) if it satisfies CSP under all conditions, in which case the ODE model corresponds to a single graph model. Typically, an ODE model corresponds to different graph models under different conditions characterizing the context-dependent and time-varying nature of biological systems [7, 8]. An ODE model that does not satisfy GCSP is illustrated in Fig 1.

One particularly rich class of ODE models that satisfy GCSP are based on GeneNetWeaver [10, 11], the software used to generate expression data in DREAM challenges 3–5 [11–13] and recently applied to single-cell analysis [14, 15]. In these ODE models a layer of intermediate elements called *modules* are constructed with transcription factors (TFs) as their input and target genes their output. The activity level of a module depends on its input and its type, and determines the production rate of its output. The modules model the binding of protein complexes to DNA in transcriptional regulation. TFs can regulate the target gene through one or multiple modules. Assuming for each TF and each target gene there is only one module that takes the TF as an input and the target gene as an output, we show that CSP is satisfied, so each GeneNetWeaver ODE model has a well-defined graph model associated with it. The combinatorial nature of the number of possible module configurations (i.e., the number of the modules and their input and output) and the continuous value parameters make the GeneNetWeaver ODE models extremely rich.

The organization of this paper is as follows. In the first subsection of the Materials and Methods section, we describe the ODE models and the graph models, and propose two notions of CSP. In the second subsection of the Materials and Methods section, we describe ODE models based on GeneNetWeaver. The Results section has three subsections. In the first, a relation of the two notions of CSP is provided. In the second, the GeneNetWeaver ODE models are shown to satisfy the constant sign property, and their complexity is investigated. In the third, a case study of a core soybean flowering network based on the literature is presented to demonstrate the use of the GeneNetWeaver ODE models. First it is illustrated that a single signed, directed graph model has a large space of consistent ODE models. Second, to study how different the GeneNetWeaver ODE models are, we explore the problem of numerically fitting parameters of one ODE model to synthetic expression data generated

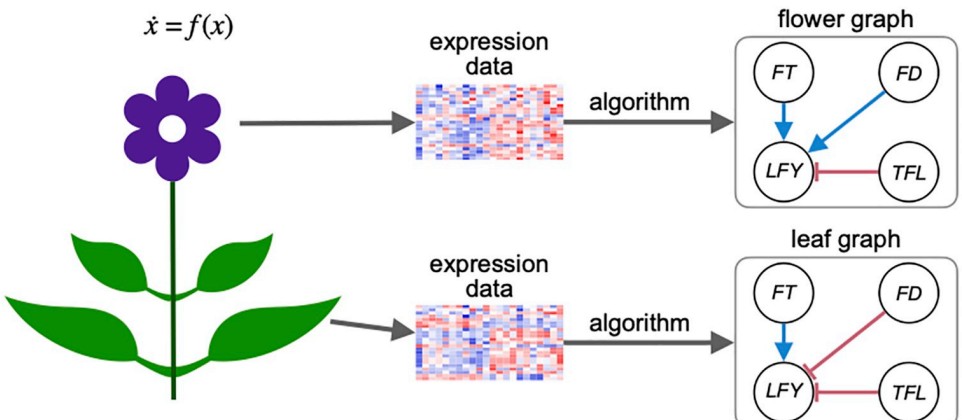

**Fig 1. Network reconstruction for an ODE model in the study [9] without global CSP.** The ODE model $f$ governs the dynamics of all parts of the plant, and expression data collected from different parts of a plant (flower vs. leaf) can correspond to different graph models.

from another. The generalization, implication and limitation of CSP are discussed before the concluding remarks.

## Materials and methods

### ODE model and constant sign property

In this section we define the constant sign property, a property under which ODE models are consistent with signed directed graphs. Roughly speaking, CSP holds when unilaterally increasing the expression level of one gene causes the expression level of another gene to move in one direction. In other words, the effect of one regulator gene has a constant sign on a target gene. In rare cases, CSP may hold globally, regardless of the expression levels of all the genes and the concentrations of any other molecular classes. More generally, CSP may hold only for a set of expression levels and system parameters, leading to a local definition. We present the precise definition of CSP in this section.

Let $x_1(t), x_2(t), \ldots, x_n(t)$ be the mRNA abundances for the $n$ genes (the observables) at time $t$. Let $x_{n+1}(t), x_{n+2}(t), \ldots, x_{n+m}(t)$ be the protein concentrations (the unobservables) at time $t$, which may include derived (protein complexes and modifications like protein phosphorylation) and localized (e.g., cytoplasmic and nuclear) proteins. Let $x_{n+m+1}(t), x_{n+m+2}(t), \ldots, x_{n+m+l}(t)$ be the strengths of the chemical and environmental signals (the controllables, e.g., temperature and photoperiod) at time $t$. Let $x(t) = (x_i(t):i\in[n+m+l])$ be the system state at time $t$, where $[n]$ denotes the set of integers $\{1, 2, \ldots, n\}$. Let $\lambda \in \mathbb{R}^s$ be the parameters of the ODE model and let $f_i : \mathbb{R}^{n+m+l} \times \mathbb{R}^s \to \mathbb{R}$ be the time derivative of $x_i$ as a function of the $(n + m + l)$-dimensional system state and the parameters for $i \in [n + m]$. Note the domain of $f_i$ is assumed to be the entire Euclidean space rather than a subset of it without loss of generality because one can always restrict $f_i$ to a subset of states that $x$ takes. Examples of $f$ for the single-input case $(n + m + l = 1)$ include the Michaelis–Menten kinetics and the more general Hill kinetics. Examples of $f$ for the multi-input case $(n + m + l \geq 2)$ include the Shea–Ackers model [16, 17], which is the average production rate based on a Gibbs measure of the control states, and the GeneNetWeaver model to be discussed later in this paper, which models the additive effect of multiple intermediate Shea–Ackers type modules. Both the Shea–Ackers model and the GeneNetWeaver model generalize the Hill kinetics to multi-input scenarios in their own ways

and are, among many other sophisticated ODE models, within the framework of ODE models in this paper.

Formally, given the numbers of molecular classes (i.e., $n$ classes of mRNAs, $m$ classes of proteins, and $l$ classes of molecular signals), the dynamics of an ODE model are characterized by the collection of time derivatives for the uncontrollable variables $f = (f_i : i \in [n + m])$. In the rest of the paper an ODE model refers to the collection of the functions $f$. The trajectories of the mRNA and protein concentrations evolving with time depend on $(x^0, \tilde{x}, \lambda)$, where $x^0 = (x_i^0 : i \in [n + m])$ are the initial conditions of the mRNAs and proteins at time 0, $\tilde{x} = (\tilde{x}_i(t) : n + m + 1 \leq i \leq n + m + l, t \geq 0)$ are the predefined external signal strengths for all time, and $\lambda \in \mathbb{R}^s$ are the parameters. The trajectories can then be obtained by solving the following initial value problem.

$$x_i(0) = x_i^0, \quad i \in [n + m],$$

$$x_i(t) = \tilde{x}_i(t), \quad n + m + 1 \leq i \leq n + m + l, t \geq 0,$$

$$\frac{dx_i(t)}{dt} = f_i(x(t), \lambda), \quad i \in [n + m].$$

Note the signals $(x_i : n + m + 1 \leq i \leq n + m + l)$ are exogenously controlled and not solved via the equations. In this paper we assume existence and uniqueness of the solution on the entire positive time horizon for ease of exposition. The concept of CSP can be easily generalized to ODE models where only local solutions exist.

**Infinitesimal monotonicity.** We first define a version of monotonicity called infinitesimal monotonicity such that CSP using this definition of monotonicity can be applied to a broad class of ODE models.

Roughly speaking, infinitesimal monotonicity characterizes the monotone influence of one observed variable on another over a sufficiently short period of time. Such monotonicity depends on the current system state. For each regulator–target pair, to avoid external and indirect influence, we clamp the exogenous signals as well as the observed variables other than the target to their initial values, so only the unobserved variables and the target observed variable are allowed to change with time. The clamped value of the regulator can be perturbed. A change in the constant value of the regulator can cause a change in the target observed variable in continuous time, possibly through one or multiple unobserved variables. The system with the input at the regulator observable and output at the target observable is thus treated as a black box in the sense that one does not need to know its internal states (the unobservables) to determine the infinitesimal monotonicity of the system. This assumes that the initial internal states are fixed.

Given the ODE model $f$, and given a state $x \in \mathbb{R}^{n+m+l}$ and parameters $\lambda \in \mathbb{R}^s$, let $j$ be the target gene and let the dynamics of the clamped ODE model be driven by

$$\hat{f}_k^{(j)} = \begin{cases} f_k & \text{if } k \in \{j\} \cup [n + 1 : n + m], \\ 0 & \text{otherwise}, \end{cases}$$

for any $k \in [n + m + l]$. Here $[a : b]$ denotes the set of integers $\{a, a + 1, \ldots, b\}$. Then $\hat{f}^{(j)} = (\hat{f}_k^{(j)} : k \in [n + m + l])$ determines the dynamics of a system where the mRNA abundances and exogenous signals remain constant across time except for the mRNA abundance of gene $j$. Fix a potential regulator gene $i \neq j$ and let $(\eta^{(j)}(t, h, x, \lambda) \in \mathbb{R}^{n+m+l} : t \geq 0)$ be the solution of the initial value problem with initial condition $(x_i + h, x_{-i})$, dynamics $\hat{f}^{(j)}$,

parameters $\lambda$. Note here $\eta^{(j)}$ also includes the clamped exogenous signals. Also note that for any $t$ we have

$$\eta_k^{(j)}(t, h, x, \lambda) \equiv x_k \text{ for } k \in [n] \setminus \{i, j\} \text{ and } k > n + m,$$

$$\eta_i^{(j)}(t, h, x, \lambda) \equiv x_i + h,$$

and

$$\eta_j^{(j)}(0, h, x, \lambda) = x_j.$$

The following definition gives a precise characterization of the target gene expression to be strictly increasing or decreasing with respect to the regulator gene expression in a small future time period.

*Definition* 1 (Infinitesimal monotonicity). For an ODE model $f$ at state $x$ with parameters $\lambda$ and $(i, j) \in [n]^2$ with $i \neq j$, the infinitesimal monotonicity for $i$ on $j$ is given by

$$B_{\mathrm{inf}}(i, j, x, \lambda) = \begin{cases} \emptyset & \text{if } \forall h \text{ and } \forall t, \eta_j^{(j)}(t, h, x, \lambda) = \eta_j^{(j)}(t, 0, x, \lambda), \\[2mm] \{1\} & \text{if } \exists \epsilon > 0 \text{ such that } \forall t \in (0, \epsilon) \text{ and } \forall h \in (-\epsilon, 0) \cup (0, \epsilon), \\[1mm] & \dfrac{\eta_j^{(j)}(t, h, x, \lambda) - \eta_j^{(j)}(t, 0, x, \lambda)}{h} > 0, \\[2mm] \{-1\} & \text{if } \exists \epsilon > 0 \text{ such that } \forall t \in (0, \epsilon) \text{ and } \forall h \in (-\epsilon, 0) \cup (0, \epsilon), \\[1mm] & \dfrac{\eta_j^{(j)}(t, h, x, \lambda) - \eta_j^{(j)}(t, 0, x, \lambda)}{h} < 0, \\[2mm] \{1, -1\} & \text{otherwise.} \end{cases}$$

Equivalently, in less mathematical terms, $B_{\mathrm{inf}}(i, j, x, \lambda) = \emptyset$ indicates gene $i$ does not affect gene $j$ at state $x$ and parameters $\lambda$. The cases with $B_{\mathrm{inf}}(i, j, x, \lambda) = \{1\}$ and $\{-1\}$ indicate gene $i$ activates or represses gene $j$, respectively, at state $x$ and parameters $\lambda$ in a small time period with small perturbation. The case with $B_{\mathrm{inf}}(i, j, x, \lambda) = \{1, -1\}$ indicates gene $i$ does not affect gene $j$ in a monotone way.

*Remark* 1. Note the case $B_{\mathrm{inf}}(i, j, x, \lambda) = \{1, -1\}$ can happen when the expression level of the target gene $j$ reaches the maximum with respect to $x_i$, so that a change of $x_i$ in either direction will cause the solution $\eta_j^{(j)}(t, h, x, \lambda)$ to decrease for small $t$, in which case the monotonicity is indeterminate (neither increasing nor decreasing).

In practice the values of $x$ and $\lambda$ may be unknown, so we are interested in how $B_{\mathrm{inf}}$ varies with $x$ and $\lambda$. Usually we expect some level of continuity of $B_{\mathrm{inf}}$ with respect to $x$ and $\lambda$, so the infinitesimal monotonicity of the ODE model may be consistent in a small set of $(x, \lambda)$ pairs, denoted by $S$. In the case when $S$ equals the entire state–parameter space, the infinitesimal monotonicity is consistent globally. The following definition generalizes Definition 1 by checking the consistency of infinitesimal monotonicity over a set $S$, and defines an associated graph.

*Definition* 2 (Infinitesimal gene regulatory graph). The infinitesimal gene regulatory graph of an ODE model $f$ over $S \subseteq \mathbb{R}^{n+m+l} \times \mathbb{R}^s$ is given by a graph $([n], \mathcal{E}_{\mathrm{inf}}(S), B_{\mathrm{inf}}(S))$, where the set of edge labels $B_{\mathrm{inf}}(S) = (B_{\mathit{inf}}(i, j, S) : (i, j) \in [n]^2, i \neq j)$ is defined by

$$B_{\mathrm{inf}}(i, j, S) = \bigcup_{(x, \lambda) \in S} B_{\mathrm{inf}}(i, j, x, \lambda)$$

and the set of edges is

$$\mathcal{E}_{\mathrm{inf}}(S) = \{(i,j) : B_{\mathrm{inf}}(i,j,S) \neq \emptyset\}.$$

Equivalently, in less mathematical terms, $B_{\mathrm{inf}}(i, j, S) = \emptyset$ indicates gene $i$ does not affect gene $j$ when $(x, \lambda)$ is in $S$. The case with $B_{\mathrm{inf}}(i, j, S) = \{1\}$ indicates gene $i$ can increase gene $j$ for some $(x, \lambda)$ in $S$, but cannot decrease gene $j$ for any $(x, \lambda)$ in $S$. The case with $B_{\mathrm{inf}}(i, j, S) = \{-1\}$ indicates gene $i$ can decrease gene $j$ for some $(x, \lambda)$ in $S$, but cannot increase gene $j$ for any $(x, \lambda)$ in $S$. The case with $B_{\mathrm{inf}}(i, j, S) = \{1, -1\}$ indicates the monotonicity is indeterminate over $S$.

*Definition* 3 (Infinitesimal constant sign property). An ODE model $f$ satisfies the infinitesimal constant sign property over $S \subseteq \mathbb{R}^{n+m+l} \times \mathbb{R}^s$ if $\forall (i, j) \in \mathcal{E}_{\mathrm{inf}}(S)$, $B_{\mathrm{inf}}(i, j, S) = \{1\}$ or $B_{\mathrm{inf}}(i, j, S) = \{-1\}$. In other words, the ODE model satisfies infinitesimal constant sign property on $S$ if no pair of $(i, j)$ has indeterminate monotonicity on $S$.

*Remark* 2. The set $S$ represents the set of states where the infinitesimal CSP holds. If $S$ is the entire state space then we say the infinitesimal CSP holds globally. Complex biological systems usually do not satisfy CSP globally, but may satisfy CSP locally over the set $S$ where the system states reside. For example, in Fig 1, the gene expressions in the flowers may be contained in set $S_1$ where the infinitesimal CSP is satisfied with a gene regulatory graph $G_1$, while the gene expressions in the leaves may be contained in set $S_2$ that does not intersect with $S_1$, and the infinitesimal CSP is satisfied with a different gene regulatory graph $G_2$.

**Sum–product monotonicity.** Infinitesimal monotonicity gives a natural notion of monotonicity, but it is expressed in terms of the solutions of the differential equations, and solving the differential equations can be analytically challenging and numerically unstable. Hence, in this section we focus on ODE models with a smooth $f$ and propose another notion of monotonicity that does not require solving the system of ODEs.

*Definition* 4 (Molecular graph). The molecular graph of an ODE model is a graph whose vertices are the internal molecular classes (i.e., the observables and the unobservables) and whose edges indicate non-constant effects among the internal molecular classes with signs indicating monotonicity of the effects. Formally, given an ODE model $f$, the molecular graph at state $x \in \mathbb{R}^{n+m+l}$ with parameters $\lambda \in \mathbb{R}^s$ is a directed graph with vertices $[n + m]$ and edges $\mathcal{E}_{\mathrm{mol}}$, where

$$\mathcal{E}_{\mathrm{mol}} = \{(i,j) \in [m+n]^2 : \text{ there exists } x \in \mathbb{R}^{n+m+l}, \lambda \in \mathbb{R}^s, \text{ and } x_i' \in \mathbb{R} \text{ such that}$$
$$f_j(x, \lambda) \neq f_j((x_i', x_{-i}), \lambda)\}.$$

In other words $(i, j) \notin \mathcal{E}_{\mathrm{mol}}$ if $f_j$ does not actually depend on $x_i$. See Fig 2(A) for an example of a molecular graph. Note in general we could have edges from unobservables to unobservables (e.g., protein–protein interactions) and from observables to observables (modeling fast translation where mRNA abundances and protein concentrations are considered the same).

The molecular graph represents the interactions among all the molecular classes. However, usually only the mRNA abundances are measured; the proteins and their derived products are not measured, making the molecular graph only partially observed. As a result, one often seeks an induced graph on the mRNA classes, which leads to the following definitions analogous to the clamped systems for infinitesimal monotonicity.

*Definition* 5 (Unobserved path of length $q$ for $q \geq 1$). Given a molecular graph, the set of unobserved paths from one mRNA to another is the set of paths that do not go though another mRNA. Formally, given $n, m, l$, and edges $\mathcal{E}_{\mathrm{mol}} \subseteq [n + m]^2$ and $i, j \in [n]$ with $i \neq j$, the set of

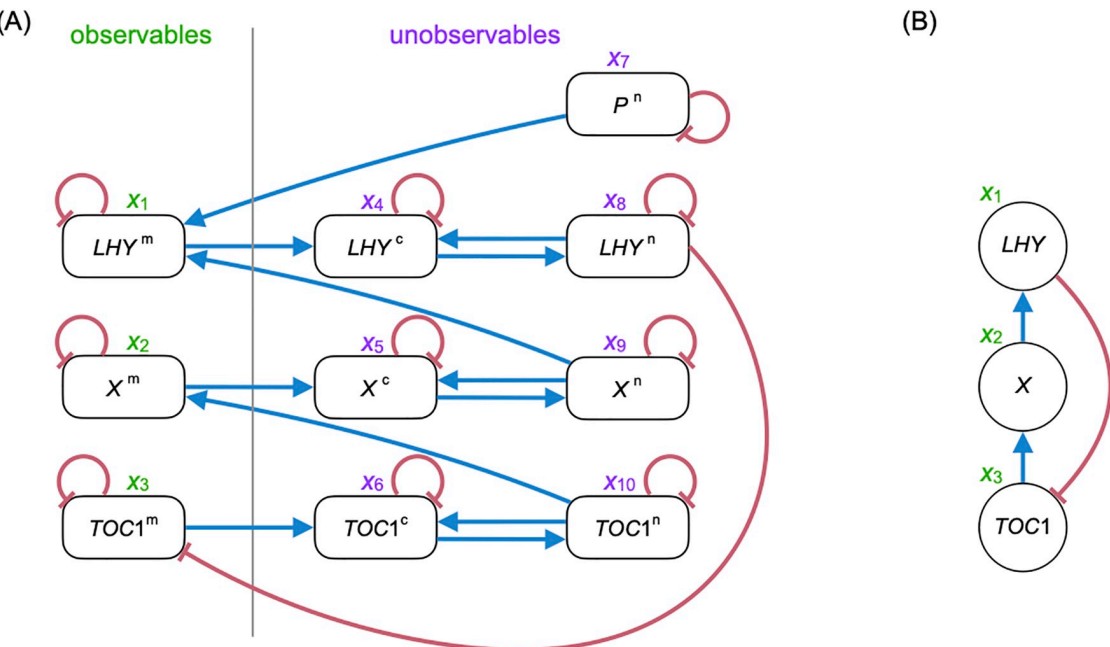

**Fig 2. A molecular graph and its corresponding gene regulatory graph for the single-loop network in the study [18].** (A) The molecular graph for the ODE model of the single-loop network. Blue edges indicate positive first-order partial derivatives, and red edges indicate negative first-order partial derivatives. (B) The corresponding global gene regulatory graph for (A) with blue edges indicating activation and red edges indicating repression (the constant sign property is satisfied globally under both notions of CSP by Proposition 1).

unobserved paths of length $q$ connecting $i$ and $j$ is

$$\mathcal{P}^q_{ij} = \Big\{ (r_0, r_1, \ldots, r_q) \in [n+m]^{q+1} : r_q = i, r_0 = j, \text{ and } \forall k \in [1:q-1], r_k \in [n+1:n+m],$$

$$\text{and } \forall k \in [q], (r_k, r_{k-1}) \in \mathcal{E}_{\mathrm{mol}} \Big\}.$$

*Definition* 6 (Molecular distance). The molecular distance from $i$ to $j$ is

$$q^*_{ij} = \begin{cases} \min\{q : \mathcal{P}^q_{ij} \neq \emptyset\} & \text{if } \mathcal{P}^q_{ij} \neq \emptyset \text{ for some } q, \\ \infty & \text{otherwise.} \end{cases}$$

*Definition* 7 (Sum–product monotonicity). For genes $i$ and $j$, state $x$ and parameters $\lambda$, the sum–product monotonicity is defined by

$$B_{\mathrm{sum}}(i, j, x, \lambda) = \begin{cases} \emptyset & \text{if } q^*_{ij} = \infty, \\ \{1\} & \text{if } q^*_{ij} < \infty \text{ and } \Delta(i, j, x, \lambda) > 0, \\ \{-1\} & \text{if } q^*_{ij} < \infty \text{ and } \Delta(i, j, x, \lambda) < 0, \\ \{1, -1\} & \text{if } q^*_{ij} < \infty \text{ and } \Delta(i, j, x, \lambda) = 0, \end{cases}$$

where $\Delta(i, j, x, \lambda) \triangleq \sum_{r \in \mathcal{P}^{q^*_{ij}}_{ij}} \prod_{l=1}^{q^*_{ij}} \partial_{r_l} f_{r_{l-1}}(x, \lambda)$.

Note $B_{\text{sum}}$ is only based on derivatives of $f$, not solving the ODEs. It plays a similar role as $B_{\text{inf}}$. Thus we can define sum–product gene regulatory graph and sum–product constant sign property in a similar way as Definitions 2 and 3. A relation between the infinitesimal monotonicity and the sum–product monotonicity is given in 1 in the Results section.

## GeneNetWeaver ODE model

We consider a differential equation model such that transcription factors participate in modules which bind to the promoter regions of a given target gene. This model is based on the GeneNetWeaver software version 3 [10]. Part of the model of the popular simulator is described in the studies [12] and [11], but there is no good reference that precisely describes the model. So in this section we describe the generative model in GeneNetWeaver based on a given directed graph, and show in the next section that the CSP is satisfied. Note GeneNet-Weaver models are a special class of ODE models with the molecular graphs being bipartite, resulting in no unobserved paths of length greater than 2, unlike the general case as illustrated in Fig 2. GeneNetWeaver allows fast protein–protein interactions though the $f$ function, but does not characterize slow protein–protein interactions or external signals.

The model in GeneNetWeaver is based on standard modeling assumptions (see [19]) including statistical thermodynamics, as described in the study [20]. The activity level of the promoter of a gene is controlled by one or more *cis*-regulatory modules, which for brevity we refer to as modules. A module can be either an enhancer or a silencer. Each module has one or more transcription factors as activators, and possibly one or more TFs as deactivators. For each target gene, a number of modules are associated with its TFs such that each TF is an input of one of the modules. For simplicity assume that each module regulates only a single target gene.

Let $([n], \mathcal{E}, b)$ be a directed signed graph with vertices $[n]$, edge set $\mathcal{E}$, and edge signs $b$. For target gene $j$, let $N_j \triangleq \{i \in [n] : (i, j) \in \mathcal{E}\}$ be the set of its TFs and let $\mathcal{S}_j \subseteq \mathcal{P}(N_j)$ be a partition of $N_j$ according to the input of the modules. Then the modules for target gene $j$ can be indexed by the tuple $(K, j)$ (denoted by $K{:}j$ in the subscripts), where $K \in \mathcal{S}_j$. Note each TF regulates the target gene $j$ only through one module. The random model for assignment of the TFs to modules and of the parameters in GeneNetWeaver is summarized in S2 Appendix. Let the sets of activators and deactivators for module $K{:}j$ be $A_{K{:}j}$ and $D_{K{:}j}$ with $A_{K{:}j} \cup D_{K{:}j} = N_j$ and $A_{K{:}j} \cap D_{K{:}j} = \emptyset$. For a module $K{:}j$, let $c_{K{:}j}$ be the type (1 for enhancer and $-1$ for silencer), $r_{K{:}j}$ the mode (1 for synergistic binding and 0 for independent binding). Note $r_{K{:}j}$ only matters for multi-input modules (i.e., those with $|K| > 1$). Let $\beta_{K{:}j} \geq 0$ be the absolute effect of module $K{:}j$ on gene $j$ in mRNA production rate. Note that by the construction in S2 Appendix, it is guaranteed that $b_{ij} = c_{K{:}j}(1_{\{i \in A_{K{:}j}\}} - 1_{\{i \in D_{K{:}j}\}})$.

Let $x_i(t)$ and $y_i(t)$ be the mRNA and protein concentrations for gene $i$ at time $t$. We ignore $t$ in the remainder of the paper for simplicity. The dynamics are given by

$$\frac{dx_i}{dt} = f_i(y) - \delta_i x_i$$

and

$$\frac{dy_i}{dt} = f_i^{(p)}(x_i) - \delta_i^{(p)} y_i,$$

where $f_i(y)$ is the relative activation rate for gene $i$ (i.e. the mRNA production rate for gene $i$ for the normalized variables) discussed in the next two subsections, $f_i^{(p)}(x_i) = \rho_i x_i$ is the translation rate of protein $i$, and $\delta_i$ and $\delta_i^{(p)}$ are the degradation rates of the mRNA and the protein.

Because only $x$ is observed in RNA-seq experiments, without loss of generality the unit of the unobserved protein concentrations can be chosen such that $\rho_i = \delta_i^{(p)}$ for all $i$ (see nondimensionalization in the study [12]). Note the GeneNetWeaver model is a special ODE model with $m = n$ and $l = 0$.

**Activity level of a single module.** For edge $(i, j)$, the normalized expression level of gene $i$, $v_{ij}$, is defined by

$$v_{ij} = \left(\frac{y_i}{k_{ij}}\right)^{h_{ij}},$$

where $k_{ij}$ is the Michaelis–Menten normalizing constant and $h_{ij}$ is a small positive integer, the Hill constant, representing the number of copies of the TF $i$ that need to bind to the promoter region of gene $j$ to activate the gene. (If gene $i$ is not bound to the promoter region of gene $j$, it is like taking the Hill constant equal to zero and thus normalized expression level equal to one.) The activity level of module $K: j$ denoted by $M_{K:\,j}$, which is the probability that module $K: j$ is active, is given in the following three cases.

**Type 1 modules: Input TFs bind to module independently**
In this case, $r_{K:\,j} = 0$, and we have

$$M_{K:j} = \left(\prod_{i \in A_{K:j}} \frac{v_{ij}}{1 + v_{ij}}\right)\left(\prod_{i \in D_{K:j}} \frac{1}{1 + v_{ij}}\right).$$

Interpreting each fraction as the probability that an activator is actively bound (or a deactivator is not bound), the activation $M_{K:\,j}$ is the probability that all the inputs of module $K: j$ are working together to activate the module, i.e., the probability that the module is active. It is assumed that for a module to be active, all the activators must be bound and all the deactivators must be unbound, and all the bindings happen independently.

One can think of module $K: j$ as a system with $2^{|A_{K:\,j}|} + |D_{K:\,j}|$ possible states of the inputs. Suppose each input $j$ binds with rate $v_{ij}$ and unbinds with rate 1 independently. Then the stationary probability of the state that all the activators are bound and none of the deactivators is bound is $M_{K:\,j}$.

Alternatively, one can assign additive energy of

$$E_{ij} = -\log v_{ij}$$
$$= -h_{ij} \log \frac{y_i}{k_{ij}}$$

to each bound input gene $i$ and energy zero to each unbound gene. Then $M_{K:\,j}$ is the probability that all activators are bound and none of the deactivators is bound in the Gibbs measure. In other words, the Type 1 modules are Shea–Ackers models with all binding states possible and only the one state with all the activators initiating transcription.

**Type 2 modules: TFs are all activators and bind to module as a complex**
In this case, $D_{K:\,j} = \emptyset$, $r_{K:\,j} = 1$, and we have

$$M_{K:j} = \frac{\prod_{i \in A_{K:j}} v_{ij}}{1 + \prod_{i \in A_{K:j}} v_{ij}}.$$

One can think of such a module as a system with only two states: bound by the activator complex, or unbound. The transition rate from unbound to bound is $\prod_{i \in A_{K:j}} v_{ij}$, and that from

bound to unbound is 1. Then the activation of the module is the probability of the bound state in the stationary distribution, given by $M_{K:\,j}$.

Alternatively, this corresponds to the Shea–Ackers model as in the previous case, except all the states other than fully unbound and fully bound are unstable (i.e. have infinite energy).

**Type 3 modules: Some TFs are deactivators and bind to module as a complex**

In this case, $D_{K:\,j} \neq \emptyset$ and $r_{K:\,j} = 1$, and we have

$$M_{K:j} = \frac{\prod_{i \in A_{K:j}} v_{ij}}{1 + \prod_{i \in A_{K:j}} v_{ij} + \left(\prod_{i \in A_{K:j}} v_{ij}\right)\left(\prod_{i \in D_{K:j}} v_{ij}\right)}. \tag{1}$$

In this case the system can be in one of three states: unbound, bound by the activator complex, and bound by the deactivated (activator) complex. The Gibbs measure in the Shea–Ackers model for Type 3 modules with three stable states (i.e. have finite energy) assigns probability $M_{K:\,j}$ to the activated state.

Note if $\prod_{i \in \emptyset} v_{ij}$ is understood to be 0 then Eq (1) reduces to Type 2 when $D_{K:\,j} = \emptyset$. However historically $\prod_{i \in D_{K:j}} v_{ij}$ was understood as 1 in an early version of GeneNetWeaver and caused a bug of wrong Type 2 modules.

*Remark* 3. Presumably it is possible for there to be more than three stable states for a module, so additional types of modules could arise, but for simplicity, following GeneNetWeaver, we assume at least one of the three cases above holds.

*Remark* 4. If a module $K$: $j$ has only one input $i$ (i.e. $K = \{i\}$) then the module is type 1 and $M_{K:j} = \frac{v_{ij}}{1+v_{ij}}$ or $M_{K:j} = \frac{1}{1+v_{ij}}$. We will see later in the random model of GeneNetWeaver that only the former (single activator) is allowed.

GeneNetWeaver software uses the 3 types of modules derived above. In all three cases the activation $M_{K:\,j}$ is monotonically increasing in $y_i$ for activators $i \in A_{K:\,j}$, and monotonically decreasing in $y_i$ for deactivators $i \in D_{K:\,j}$.

**Production rate as a function of multiple module activations.** The relative activation of gene $j$ as a function of the protein concentrations $y$ is

$$f_j(y) = \sum_{s \in \{0,1\}^{\mathcal{S}_j}} \alpha_{j,s} \left(\prod_{K \in \mathcal{S}_j : s_K = 1} M_{K:j}\right)\left(\prod_{K \in \mathcal{S}_j : s_K = 0} (1 - M_{K:j})\right), \tag{2}$$

where $\alpha_{j,s}$ is the relative activation of the promoter under the module configuration $s$. Note that $\alpha$ in Eq (2) gives $2^{|\mathcal{S}_j|}$ degrees of freedom, one for every possible subset of the modules being active. However, following the GeneNetWeaver computer code [10], we assume that the interaction among the modules is linear, meaning that for some choice of $\alpha_{j,\text{basal}}$, $(c_{K:j} : K \in \mathcal{S}_j)$, and $(\beta_{K:j} : K \in \mathcal{S}_j)$, we have for any configuration $s \in \{0,1\}^{\mathcal{S}_j}$,

$$\alpha_{j,s} = \alpha_{j,\text{basal}} + \sum_{K \in \mathcal{S}_j : s_K = 1} c_{K:j} \beta_{K:j}, \tag{3}$$

This reduces the number of degrees of freedom for $\alpha$ to $|\mathcal{S}_j| + 1$. Then, combining Eqs (2) and (3) yields

$$\begin{aligned}
f_j(y) &= \mathbb{E}\alpha_{j,S} \\
&= \alpha_{j,\text{basal}} + \sum_{K \in \mathcal{S}_j} c_{K:j} \beta_{K:j} \mathbb{E}S_K \\
&= \alpha_{j,\text{basal}} + \sum_{K \in \mathcal{S}_j} c_{K:j} \beta_{K:j} M_{K:j},
\end{aligned} \tag{4}$$

where $S$ is distributed by the product distribution of the Bernoulli distributions with means $(M_{K:j} : K \in \mathcal{S}_j)$. So the relative activation, or the mRNA production rate, of a gene is given by the basal activation plus the inner product of the module effects and the module activation. We also note that the effect of the modules is not assumed to be statistically independent: all we need to know to compute the relative activation of a gene are the marginal probability of activation of the single modules.

Taking into account the three different types of modules described in the previous section on activity level of a single module, Eq (4) yields the following expression for the relative activation of gene $j$:

$$
\begin{aligned}
f_j(y) =\ & \alpha_{j,\text{basal}} + \sum_{K:r_{K:j}=0} c_{K:j}\beta_{K:j} \left( \prod_{i \in A_{K:j}} \frac{v_{ij}}{1+v_{ij}} \right) \left( \prod_{i \in D_{K:j}} \frac{1}{1+v_{ij}} \right) \\
& + \sum_{\substack{K\,:\,r_{k_j}=1 \\ D_{K:j}=\emptyset}} c_{K:j}\beta_{K:j} \frac{\prod_{i \in A_{K:j}} v_{ij}}{1+\prod_{i \in A_{K:j}} v_{ij}} \\
& + \sum_{\substack{K\,:\,r_{k_j}=1 \\ D_{K:j}=\emptyset}} c_{K:j}\beta_{K:j} \frac{\prod_{i \in A_{K:j}} v_{ij}}{1+\prod_{i \in A_{K:j}} v_{ij} + \left( \prod_{i \in A_{K:j}} v_{ij} \right)\left( \prod_{i \in D_{K:j}} v_{ij} \right)}.
\end{aligned}
\tag{5}
$$

As we will see in the Results section, $f$ satisfies the CSP. Note that in the actual GeneNet-Weaver source code every $\alpha_{j,s}$ is truncated to the interval $[0, 1]$:

$$
\alpha_{j,s} = \left[ \alpha_{j,\text{basal}} + \sum_{K \in \mathcal{S}_j\,:\,s_K=1} c_{K:j}\beta_{K:j} \right]_0^1,
$$

where $[x]_0^1 = \max\{\min\{x, 1\}, 0\}$ is the projection of $x$ to the $[0, 1]$ interval. Then the relative activation in each state may not be linear in the individual module effects. In that case one has to resort to Eq (2) instead of Eq (5) for computing the mRNA production rate. The resulting truncated model does not necessarily satisfy the CSP because $f_j$ may not be monotone in $M_{K:j}$ in Eq (2).

## Results

### A relation between infinitesimal monotonicity and sum–product monotonicity

The following result establishes the equivalence of the two notions of monotonicity for ODE models that satisfy the sum–product CSP. So if the sum–product CSP holds, we do not need to distinguish between the sum–product CSP and the infinitesimal CSP. Consequently, given an ODE model, one can easily find the corresponding graph models for different system parameters, external signals, and internal states by calculating the sum products of the first-order partial derivatives of the input function $f$.

**Proposition 1**. *If $f$ is smooth and satisfies the sum–product CSP over $S \subseteq \mathbb{R}^{n+m+l} \times \mathbb{R}^s$, then it also satisfies the infinitesimal CSP over S, and the sum–product gene regulatory graph and the infinitesimal gene regulatory graph are the same.*

*proof*. It suffices to show $B_{sum}(i, j, x, \lambda) = B_{\inf}(i, j, x, \lambda)$ if $B_{sum}(i, j, x, \lambda) \neq \{1, -1\}$ for any $(x, \lambda) \in S$. For fixed $i, j, x, \lambda$, let $\eta(t, h) \triangleq \eta^{(j)}(t, h, x, \lambda)$ be the solution of the clamped initial value

problem at time $t$ with initial condition $\eta(0, h) = (x_i + h, x_{-i})$. We are interested in the sign of

$$g(t, h) \triangleq \eta_j(t, h) - \eta_j(t, 0).$$

If $q_{ij}^* = \infty$ then we readily have $B_{\text{sum}}(i, j, x, \lambda) = B_{inf}(i, j, x, \lambda) = \emptyset$. Suppose $q_{ij}^* = q < \infty$. Then by Corollary 4.1 in Section 5 of [21] (page 101), $f$ being smooth implies $g$ is also smooth, and we can show that (see the proof in S3 Appendix)

$$\partial_{t^a h^b} g(0, 0) = \begin{cases} \Delta(i, j, x, \lambda) & \text{if } (a, b) = (q, 1), \\ 0 & \text{if } 0 \leq a \leq q - 1 \text{ or } b = 0. \end{cases} \tag{6}$$

Hence by the multivariate Taylor's theorem (see, e.g., [22])

$$\begin{aligned} g(t, h) &= g(0, 0) + g'(0, 0)(t, h) + \frac{1}{2} g^{(2)}(0, 0)(t, h)^2 + \dots \\ &\quad + \frac{1}{(q + 1)!} g^{(q+1)}(0, 0)(t, h)^{q+1} + o(|t|^{q+1} + |h|^{q+1}) \\ &= 0 + 0 + \dots + 0 + \frac{1}{(q + 1)!} \left( \frac{\partial^{q+1} g}{\partial t^{q+1}}(0, 0) t^{q+1} + \binom{q + 1}{1} \frac{\partial^{q+1} g}{\partial t^q \partial h}(0, 0) t^q h \right. \\ &\quad \left. + \dots + \frac{\partial^{q+1} g}{\partial h^{q+1}}(0, 0) h^{q+1} \right) + o(|t|^{q+1} + |h|^{q+1}) \\ &= \frac{1}{q!} \Delta(i, j, x, \lambda) t^q h + o(|t|^{q+1} + |h|^{q+1}) \end{aligned}$$

as $(t, h) \to (0, 0)$. So $g(t, h)$ has the same sign as $\Delta(i, j, x, \lambda) t^q h$ in a sufficiently small neighborhood of $(0, 0)$. Hence $B_{\text{sum}}(i, j, x, \lambda) = B_{\text{inf}}(i, j, x, \lambda)$.

*Remark* 5. If multiple ODE models satisfy CSP with the same gene regulatory graph, then they can be combined into a single ODE model with different parameterization so that the combined ODE model still satisfies CSP with the same gene regulatory graph. For example, ODE models for different environmental temperatures can be either considered different models or a single unified model with different temperature parameter. Then the temperature-specific models satisfy CSP with the same gene regulatory graph if and only if the unified model satisfies CSP for all temperatures.

*Remark* 6. The effect of a gene on itself can be either autoregulation or degradation. The two effects can be distinguished with the molecular graph: a self-loop with negative derivative indicates degradation, and a loop of multiple hops indicates autoregulation. The infinitesimal monotonicity does not distinguish the two effects.

The following is an example of an ODE model that does not satisfy CSP globally, based on the interactions among *FT*, *TFL1*, *FD*, and *LFY* genes in the study [9].

*Example* 1. Consider a four-gene ODE model with the following dynamics for gene 4.

$$\begin{aligned} \dot{x}_4 &= f_4(x_1, x_2, x_3) \\ &\triangleq \frac{x_1 x_3}{\lambda_1 + x_1 x_3} \frac{\lambda_2}{\lambda_2 + x_2 x_3}, \end{aligned}$$

where we use $x$ for both the mRNA and protein concentrations. The biological meaning could be genes 1 and 3 form a protein complex that activates gene 4, while genes 2 and 3 form a protein complex that represses gene 4. Then it can be checked that the effect of gene 3 on gene 4

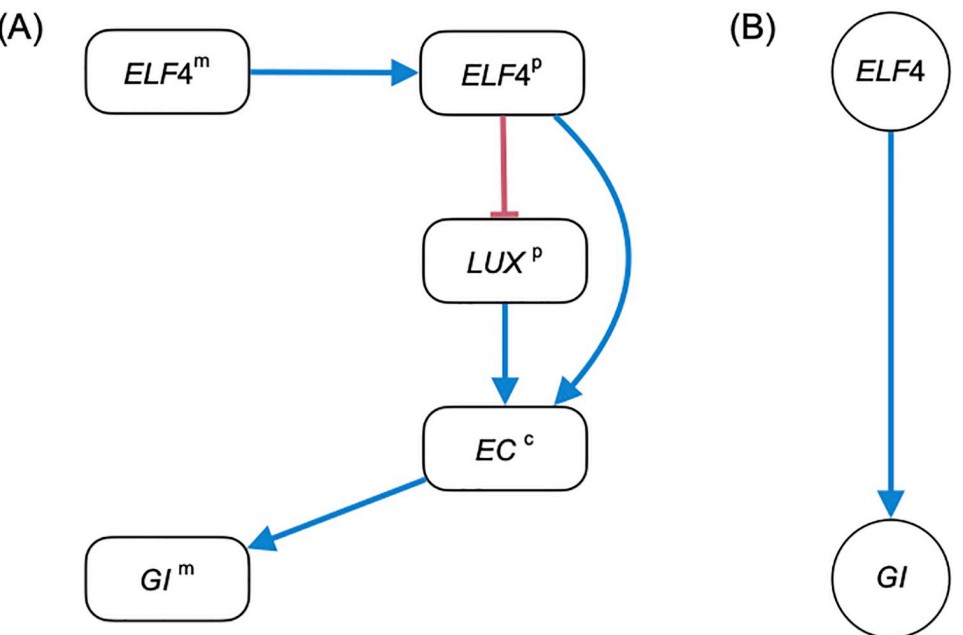

**Fig 3. Molecular graph and gene regulatory graph of the *ELF4–GI* regulation in the study [23].** (A) The molecular graph with blue edges indicating positive partial derivatives and red edges indicating negative partial derivatives. (B) The gene regulatory graph.

does not satisfy the CSP globally. Indeed, one can check that

$$\partial_3 f_4 = \frac{x_1 \lambda_2}{(\lambda_1 + x_1 x_3)^2 (\lambda_2 + x_2 x_3)^2}(\lambda_1 \lambda_2 - x_1 x_2 x_3^2).$$

So gene 3 activates gene 4 if $\lambda_1 \lambda_2 > x_1 x_2 x_3^2$, and represses gene 4 if $\lambda_1 \lambda_2 < x_1 x_2 x_3^2$.

Here is an example of a molecular graph having a shorter unobserved path dominating a longer unobserved path with the opposite sign, taken from part of the gene regulatory network in the study [23], achieving CSP with the sign of the shorter path (see Fig 3).

*Example* 2. The mRNA *ELF4*$^m$ is transcribed into the protein *ELF4*$^p$, which then forms the complex *EC*$^c$ with the protein *LUX*$^p$. The complex *EC*$^c$ induces the transcription of the mRNA *GI*$^m$. Then there is a 3-hop path (*ELF4*$^m$–*ELF4*$^p$–*EC*$^c$–*GI*$^m$) and a 4-hop path (*ELF4*$^m$–*ELF4*$^p$–*LUX*$^p$–*EC*$^c$–*GI*$^m$) from *ELF4*$^m$ to *GI*$^m$ with opposite signs. The ODE model of the molecular graph satisfies CSP with *ELF4* activating *GI* in the gene regulatory graph.

## GeneNetWeaver: CSP and complexity

In this section GeneNetWeaver models (without the truncation of the $\alpha$ terms in the implementation) are shown to satisfy the CSP globally, regardless of the parameters and the system states, and thus correspond to the signed directed graphs that were used to generate the models. Moreover, when data is generated through multifactorial perturbation for the DREAM challenge (primarily for generation of stationary expression levels, rather than trajectories), each ensemble of networks produced is also associated with the same directed signed graph. This is in contrast to the Shea–Ackers model, which is shown to be able to generate non-monotone behavior [17]. Formally we have the following result.

**Proposition** 2. *Given any directed signed graph, the ensemble of the GeneNetWeaver models satisfy CSP over* $(0, \infty)^{2n}$ *and the gene regulatory graphs coincide with the given graph.*

*Proof.* Fix any model of the ensemble of GeneNetWeaver models for the given graph. For any target gene $j$ and its regulator $i \in N_j$, there exists a unique module, indexed by $K{:}j$, whose input $K \in \mathcal{S}_j$ includes $i$. Then for any of the three module types,

$$\partial_{v_{ij}} M_{K{:}j} \begin{cases} > 0 & \text{if } i \in A_{K{:}j}, \\ < 0 & \text{if } i \in D_{K{:}j}. \end{cases}$$

Then by Eq (4),

$$\partial_{y_i} f_j = c_{K{:}j} \beta_{K{:}j} \partial_{v_{ij}} M_{K{:}j} h_{ij} \frac{y_i^{h_{ij}-1}}{k_{ij}^{h_{ij}}}$$

and

$$\partial_{x_i} f_i^{(\mathrm{p})} = \rho_i.$$

Because only $c_{K{:}j}$ and $\partial_{v_{ij}} M_{K{:}j}$ can be negative in $\partial_{y_i} f_j \partial_{x_i} f_k^{(\mathrm{p})}$, the sum–product of the first-order partial derivatives of the path from $x_i$ to $x_j$ has the same sign as $c_{K{:}j} \partial_{v_{ij}} M_{K{:}j}$, which is consistent with the sign $b_{ij}$ in the given graph by the construction in S2 Appendix. Hence by Proposition 1 the fixed ODE model satisfies CSP over all positive state vectors with gene regulatory graph equal to the given graph. Repeat this for all ODE models in the ensemble and the proposition is proved. We now discuss the complexity of GeneNetWeaver ODE models for a given gene regulatory graph. The complexity comes from both the large number of parameters and the combinatorial nature of the module configurations. The complexity indicates that ODE models are both much more detailed and considerably harder to infer compared to the graphical models.

For each gene $i$ there are 5 non-negative real parameters ($\alpha_{i,\mathrm{basal}}$, $x_i(0)$, $y_i(0)$, $\delta_i^{(\mathrm{m})}$, $\delta_i^{(p)}$). For each edge $(i, j)$ there is a non-negative real parameter ($k_{ij}$) and an integer parameter ($h_{ij}$). For each module $K{:}i$ there is a positive real parameter ($\beta_{K{:}i}$) and two binary parameters ($c_{K{:}i}$ and $r_{K{:}i}$).

The module configuration encodes great combinatorial complexity. Given a gene has $K \geq 1$ input genes, the number of ways to partition the genes into modules is the $K$th Bell number. The first ten Bell numbers are 1, 2, 5, 15, 52, 203, 877, 4140, 21147, and 115975. In addition, each input to a given module needs to be classified as an activator or deactivator.

## Case study: Soybean flowering networks

In this section the similarities of the ODE models corresponding to three different graph models are studied. First the classes of ODE models are listed for the three graph models. Then, to investigate their similarities, we generate expression data from one ODE model, and fit another model to the data by optimizing the parameters. The level of fitness of one class of ODE model to the data generated from another is used as a metric of similarity. As we will see, ODE models corresponding to the same graph model tend to have a higher similarity, while those from different graph models tend to have a lower similarity, as long as the least-squares problem is sufficiently overdetermined. The result implies that the graph model corresponding to the ODE model may be recovered with moderate amount of data, while the amount of data required for ODE model recovery may be of a much higher order. The simulation code for the data fitting results is available at [24].

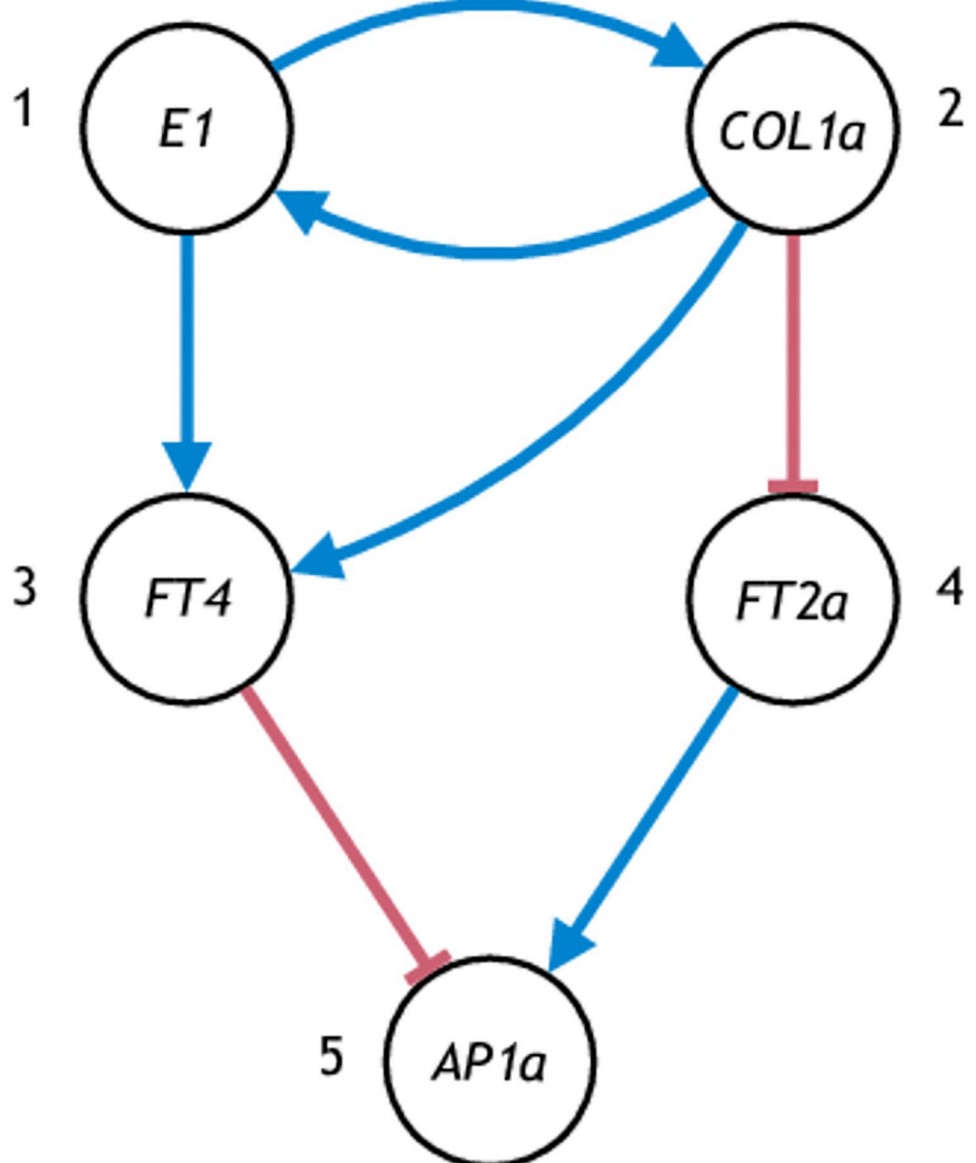

**Fig 4. A graph model of the core flowering network for soybean.**

**Five-gene graph and ODE models.** In this section we explicitly write out the classes of GeneNetWeaver ODE models of three graph models. The first two graph models are compiled from the literature, with only the sign of one edge different between them (the difference is discovered in the study [25]). The third graph model is an arbitrary five-gene repressilator for comparison purpose.

**Flowering network with *COL1a* activating *E1*.** A graph model of a five-gene soybean flowering network is shown in Fig 4. The network is based on the flowering network for *Arabidopsis* and homologs of *Arabidopsis* genes found in soybean (see Table 1). The corresponding gene IDs are shown in Table 2.

The mRNA and proteins concentrations of the soybean genes *E1*, *COL1a*, *FT4*, *FT2a*, and *AP1a* are denoted by $(x_i)_{1 \leq i \leq 5}$ and $(y_i)_{1 \leq i \leq 5}$. The differential equations based on the

**Table 1. Core flowering genes.**

| regulatory interaction | reference |
|---|---|
| *E1* activates *COL1a* | [26] |
| *E1* activates *FT4* | [27] |
| *COL1a* activates *E1* | [25] |
| *COL1a* represses *E1* | [26] |
| *COL1a* activates *FT4* | [26], [25] |
| *COL1a* represses *FT2a* | [26], [25] |
| *FT4* represses *AP1a* | [27]* |
| *FT2a* activates *AP1a* | [28] |

* For *FT4* only, not for the interaction with *AP1a*.

**Table 2. Core flowering genes.**

| index | gene ID | gene name |
|---|---|---|
| 1 | Glyma.06G207800 | *E1* |
| 2 | Glyma.08G255200 | *COL1a* |
| 3 | Glyma.08G363100 | *FT4* |
| 4 | Glyma.16G150700 | *FT2a* |
| 5 | Glyma.16G091300 | *AP1a* |

GeneNetWeaver model are

$$\dot{x}_1 = \alpha_{1,\text{basal}} + \frac{(y_2/k_{21})^{h_{21}}}{1 + (y_2/k_{21})^{h_{21}}} \beta_{2:1} - \delta_1^{(\text{m})} x_1. \tag{7}$$

$$\dot{x}_2 = \alpha_{2,\text{basal}} + \frac{(y_1/k_{12})^{h_{12}}}{1 + (y_1/k_{12})^{h_{12}}} \beta_{1:2} - \delta_2^{(\text{m})} x_2. \tag{8}$$

$$\begin{cases} \dot{x}_3 = \alpha_{3,\text{basal}} + \frac{(y_1/k_{13})^{h_{13}}}{1+(y_1/k_{13})^{h_{13}}} \frac{(y_2/k_{23})^{h_{23}}}{1 + (y_2/k_{23})^{h_{23}}} \beta_{12:3} - \delta_3^{(\text{m})} x_3 \\ \qquad\qquad\qquad\qquad\qquad \text{(independent binding), or} \\ \dot{x}_3 = \alpha_{3,\text{basal}} + \frac{(y_1/k_{13})^{h_{13}}(y_2/k_{23})^{h_{23}}}{1+(y_1/k_{13})^{h_{13}}(y_2/k_{23})^{h_{23}}} \beta_{12:3} - \delta_3^{(\text{m})} x_3 \\ \qquad\qquad\qquad\qquad\qquad \text{(synergistic binding), or} \\ \dot{x}_3 = \alpha_{3,\text{basal}} + \frac{(y_1/k_{13})^{h_{13}}}{1+(y_1/k_{13})^{h_{13}}} \beta_{1:3} + \frac{(y_2/k_{23})^{h_{23}}}{1+(y_2/k_{23})^{h_{23}}} \beta_{2:3} - \delta_3^{(\text{m})} x_3 \\ \qquad\qquad\qquad\qquad\qquad \text{(two modules).} \end{cases} \tag{9}$$

$$\dot{x}_4 = \left( \alpha_{4,\text{basal}} - \frac{(y_2/k_{24})^{h_{24}}}{1 + (y_2/k_{24})^{h_{24}}} \beta_{2:4} \right)^+ - \delta_4^{(\text{m})} x_4. \tag{10}$$

$$
\begin{cases}
\dot{x}_5 = \alpha_{5,\text{basal}} + \frac{1}{1+(y_3/k_{35})^{h_{35}}} \frac{(y_4/k_{45})^{h_{45}}}{1+(y_4/k_{45})^{h_{45}}} \beta_{34:5} - \delta_5^{(m)} x_5 \\
\qquad\qquad\qquad\text{(independent binding enhancer), or} \\[2mm]
\dot{x}_5 = \left( \alpha_{5,\text{basal}} - \frac{(y_3/k_{35})^{h_{35}}}{1+(y_3/k_{35})^{h_{35}}} \frac{1}{1+(y_4/k_{45})^{h_{45}}} \beta_{34:5} \right)^+ - \delta_5^{(m)} x_5 \\
\qquad\qquad\qquad\text{(independent binding silencer), or} \\[2mm]
\dot{x}_5 = \alpha_{5,\text{basal}} + \frac{(y_4/k_{45})^{h_{45}}}{1+(y_4/k_{45})^{h_{45}}+(y_3/k_{35})^{h_{35}}(y_4/k_{45})^{h_{45}}} \beta_{34:5} - \delta_5^{(m)} x_5 \\
\qquad\qquad\qquad\text{(synergistic binding enhancer), or} \\[2mm]
\dot{x}_5 = \left( \alpha_{5,\text{basal}} - \frac{(y_3/k_{35})^{h_{35}}}{1+(y_3/k_{35})^{h_{35}}+(y_3/k_{35})^{h_{35}}(y_4/k_{45})^{h_{45}}} \beta_{34:5} \right)^+ - \delta_5^{(m)} x_5 \\
\qquad\qquad\qquad\text{(synergistic binding silencer), or} \\[2mm]
\dot{x}_5 = \left( \alpha_{5,\text{basal}} - \frac{(y_3/k_{35})^{h_{35}}}{1+(y_3/k_{35})^{h_{35}}} \beta_{3:5} + \frac{(y_4/k_{45})^{h_{45}}}{1+(y_4/k_{45})^{h_{45}}} \beta_{4:5} \right)^+ - \delta_5^{(m)} x_5 \\
\qquad\qquad\qquad\text{(two modules).}
\end{cases}
\tag{11}
$$

$$
\dot{y}_1 = \rho_1(x_1 - y_1). \tag{12}
$$

$$
\dot{y}_2 = \rho_2(x_2 - y_2). \tag{13}
$$

$$
\dot{y}_3 = \rho_3(x_3 - y_3). \tag{14}
$$

$$
\dot{y}_4 = \rho_4(x_4 - y_4). \tag{15}
$$

$$
\dot{y}_5 = \rho_5(x_5 - y_5). \tag{16}
$$

Here $(x)^+ = \max\{x, 0\}$. We apply nondimensionalization by setting $\delta_i = \alpha_{i,\text{basal}} + \Sigma_j \beta_{j:\,i}$, so that the steady state expression levels are between 0 and 1. We can see that given the graph, there are 15 configurations of the ODEs (3 for $x_3$ times 5 for $x_5$). We use $[i, j]$ with $1 \le i \le 3$ and $1 \le j \le 5$ to denote the configuration using the $i$th equation for $x_3$ and the $j$th equation for $x_5$, and use the symbol $\text{F}_{[i,j],+}$ to denote the class of flowering network ODE models with configurations $[i, j]$ (the plus sign signifies the activation regulation of *COL1a* on *E1*). The initial conditions, namely the 5 mRNA abundances $x(0)$'s and the 5 protein concentrations $y(0)$'s, are 10-dimensional. In addition, there are 24–26 positive real parameters (depending on the configuration) and 7 discrete parameters (the Hill coefficients) for the dynamics. For example, for configuration [1, 1], the parameters for the dynamics consist of the basal activations $\alpha$'s (5), the Michaelis–Menten constants $k$'s (7), the absolute effect of modules $\beta$'s (7), the translation rate $\rho$'s (5), summing up to 24 parameters.

**Flowering network with *COL1a* repressing *E1*.**   A slight variant of the soybean flowering graph model in Fig 4 is shown in Fig 5. Note the only difference is the sign of the edge from *COL1a* to *E1*. The symbol $\text{F}_{[i,j],-}$ denotes the class of ODE models Eqs (7)–(16) with the $i$th

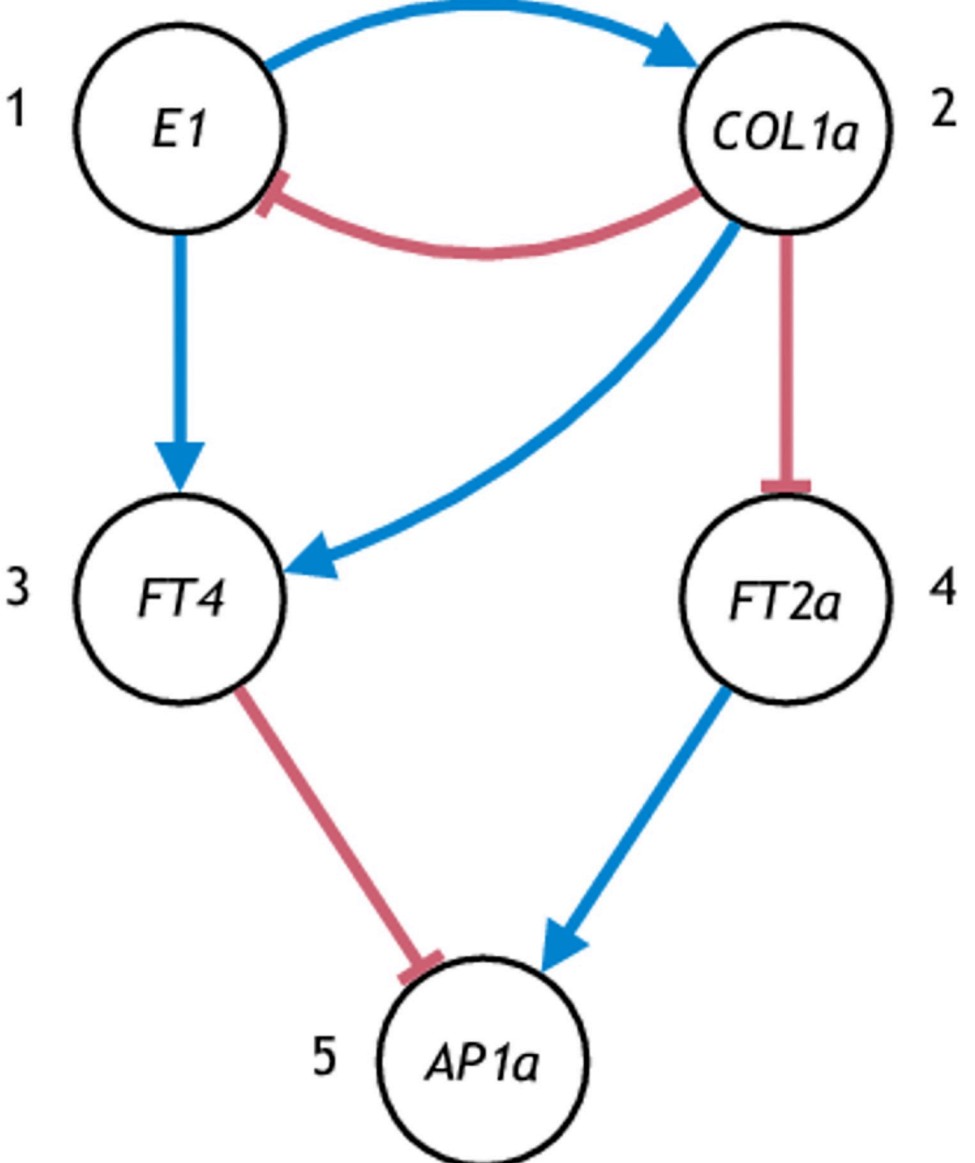

**Fig 5. A variant of the graph model of the core flowering network for soybean.**

and the $j$th configurations in Eqs (9) and (11), but with Eq (4) replaced by

$$\dot{x}_1 = \left( \alpha_{1,\text{basal}} - \frac{(y_2/k_{21})^{h_{21}}}{1 + (y_2/k_{21})^{h_{21}}} \beta_{2:1} \right)^+ - \delta_1^{(\text{m})} x_1. \tag{17}$$

Here the negative sign in $F_{[i,j],-}$ signifies the repression regulation of *COL1a* on *E1*. The number of parameters is the same as the network in Fig 4.

**Repressilator.** An arbitrary repressilator network is shown in Fig 6.

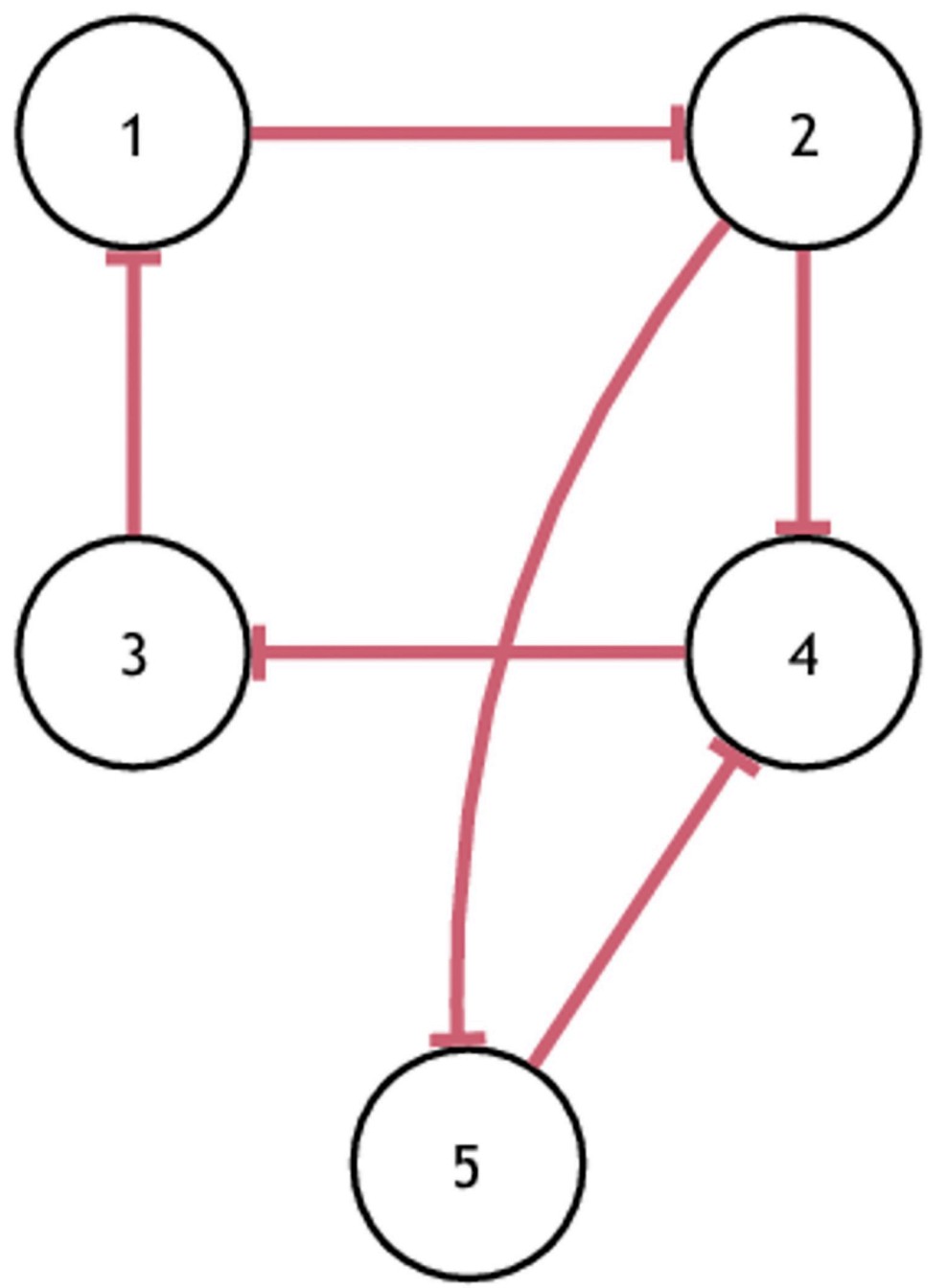

**Fig 6. A five-gene repressilator graph model.**

The symbol R denotes the class of ODE models for the repressilator, given below.

$$\dot{x}_1 = \left( \alpha_{1,\text{basal}} - \frac{(y_3/k_{31})^{h_{31}}}{1 + (y_3/k_{31})^{h_{31}}} \beta_{3:1} \right)^{+} - \delta_1^{(\text{m})} x_1. \tag{18}$$

$$\dot{x}_2 = \left( \alpha_{2,\text{basal}} - \frac{(y_1/k_{12})^{h_{12}}}{1 + (y_1/k_{12})^{h_{12}}} \beta_{1:2} \right)^{+} - \delta_2^{(\text{m})} x_2. \tag{19}$$

$$\dot{x}_3 = \left( \alpha_{3,\text{basal}} - \frac{(y_4/k_{43})^{h_{43}}}{1 + (y_4/k_{43})^{h_{43}}} \beta_{4:3} \right)^+ - \delta_3^{(\text{m})} x_3. \tag{20}$$

$$\dot{x}_4 = \left( \alpha_{4,\text{basal}} - \frac{(y_5/k_{54})^{h_{54}}}{1 + (y_5/k_{54})^{h_{54}}} \beta_{5:4} \right)^+ - \delta_4^{(\text{m})} x_4. \tag{21}$$

$$\dot{x}_5 = \left( \alpha_{5,\text{basal}} - \frac{(y_2/k_{25})^{h_{25}}}{1 + (y_2/k_{25})^{h_{25}}} \beta_{2:5} \right)^+ - \delta_5^{(\text{m})} x_5. \tag{22}$$

There is only one possible configuration for each target gene. The dynamics involve 20 parameters.

**Data generation.** The synthetic expression dataset is generated as follows. For the generated data, we use $F_{[1,1],+}$ (the flowering network with configuration [1, 1] and *COL1a* activating *E1*) with a fixed set of parameters for the dynamics. For a single set of trajectories (i.e., for a single plant), we use a set of initial values $x(0)$'s and $y(0)$'s generated uniformly at random between 0 and 1. The entire dataset may consist of only a single set of trajectories, corresponding to a single plant; or the dataset may consist of multiple sets of trajectories, corresponding to multiple plants. If multiple sets of trajectories are used, the initial conditions for each set of trajectories are generated independently, while the parameters for the dynamics are the same across all sets of trajectories. In other words, we model distinct plants by assuming distinct initial conditions, while using common parameters for the dynamics. To produce the data, the $x$ variables are sampled at time points 0, 1, 2, 3, 4, 5, 6, so that each set of trajectories (i.e., each plant) produces 35 data points. Because each set of trajectories is sampled at different times from the system with one initial condition representing different stages of a single plant, the synthetic datasets are of multi-shot sampling, as opposed to one-shot sampling in practice where each individual is only sampled once [29]. We also generate random expression datasets with reflected Brownian motions with covariance 0.05, and denote such a stochastic model by B.

**Fitting results.** The counts for data points and parameters are summarized in Table 3. Note that with a single set of trajectories, the number of parameters is close to the number of data points. As the number of sets of trajectories increases, the number of data points outgrows the number of parameters because each additional set provides 35 new data points while only allowing 10 more parameters from the initial conditions (because the dynamic parameters are shared across all sets of trajectories).

A Basin-hopping algorithm in the Python package LMFIT [30] is used to perform the global optimization of the curve fitting (see details in the source code of the simulation [24]).

**Table 3. Number of parameters in different ODE models.**

| S (number of sets of trajectories) | 1 | 2 | 5 | 10 |
|---|---|---|---|---|
| STn (number of data points) | 35 | 70 | 175 | 350 |
| $F_{[1,1],+}$ | 34 | 44 | 74 | 124 |
| $F_{[3,5],+}$ | 36 | 46 | 76 | 126 |
| $F_{[1,1],-}$ | 34 | 44 | 74 | 124 |
| R | 30 | 40 | 70 | 120 |

**Table 4. Fitting losses using different classes of ODE models on different synthetic datasets.**

| S (number of sets of trajectories) | 1 | 2 | 5 | 10 |
|---|---|---|---|---|
| fit $F_{[1,1],+}$ model to $F_{[1,1],+}$ data | 0.0015 | 0.0015 | 0.0010 | 0.0009 |
| fit $F_{[3,5],+}$ model to $F_{[1,1],+}$ data | 0.0016 | 0.0021 | 0.0019 | 0.0021 |
| fit $F_{[1,1],-}$ model to $F_{[1,1],+}$ data | 0.0032 | 0.0036 | 0.0165 | 0.0208 |
| fit $R$ model to $F_{[1,1],+}$ data | 0.0030 | 0.0037 | 0.0148 | 0.0204 |
| fit $F_{[1,1],+}$ model to $B$ data | 0.1269 | 0.1125 | 0.1307 | 0.1390 |

**Table 5. Coefficients of determination using different classes of ODE models on different synthetic datasets.**

| S (number of sets of trajectories) | 1 | 2 | 5 | 10 |
|---|---|---|---|---|
| fit $F_{[1,1],+}$ model to $F_{[1,1],+}$ data | 0.99996 | 0.99995 | 0.99999 | 0.99999 |
| fit $F_{[3,5],+}$ model to $F_{[1,1],+}$ data | 0.99995 | 0.99991 | 0.99996 | 0.99995 |
| fit $F_{[1,1],-}$ model to $F_{[1,1],+}$ data | 0.99980 | 0.99974 | 0.99702 | 0.99517 |
| fit $R$ model to $F_{[1,1],+}$ data | 0.99983 | 0.99972 | 0.99760 | 0.99535 |
| fit $F_{[1,1],+}$ model to $B$ data | 0.88639 | 0.90175 | 0.87241 | 0.87517 |

The sample size varies between 35 and 350 depending on the number of sets of trajectories. The fit is evaluated by the fitting loss and the coefficients of determination ($R^2$) shown in Tables 4 and 5. The fitting loss function for two $S \times T \times n$ tensors $x$ and $\hat{x}$ is defined by

$$l(x, \hat{x}) = \left( \frac{1}{STn} \sum_{i=1}^{S} \sum_{j=1}^{T} \sum_{k=1}^{n} (x_{ijk} - \hat{x}_{ijk})^2 \right)^{1/2},$$

where $S$ is the number of sets of trajectories in the dataset, $T$ the number of time points, and $n$ the number of genes.

Note the time scale of the ODE is assumed to be known, which restricts how fast the expression levels can change. The time scale thus acts as a regularizer to prevent overfitting.

We make the following observations from Tables 4 and 5.

1. The implemented optimization algorithm failed to find the optimal parameters in row 1 (the best fit should be a perfect fit with zero loss), but the relative loss compared to the average nondimensionalized expression level 0.5 is very small (less than 0.5%), and the coefficients of determination are close to 1. Both indicate a near-optimal fit.

2. ODE models from all three graph models (rows 1, 2, 3, and 4) fit the synthetic flowering network data well when there are only one or two sets of trajectories (columns 1 and 2). The relative losses are less than 1% and $R^2$ is larger than 0.9997. We can see from Table 3 that the number of data points is close to the number of parameters in the $S = 1$ setting, and only moderately larger in the $S = 2$ setting. So when $S \leq 2$ the three graph models in this case study are nearly indistinguishable. In other words, one may not be able to infer the graph structure with very limited data.

3. When fitting the models to 5 or 10 sets of trajectories simultaneously, i.e., when the system is sufficiently overdetermined, only the models from the correct graph (rows 1 and 2) fit well. The models from incorrect graphs (rows 3 and 4) suffer a roughly 4% relative loss after fitting for 10 sets of trajectories and $R^2$ falls below 0.998. Note that $F_{[1,1],-}$ differs from the ground truth of the data $F_{[1,1],+}$ only by the sign of one edge, while the model $R$ shares no

edges in common with the ground truth at all. Yet the fitness of the slight variant of the ground truth graph is as bad as the completely different repressilator graph.

4. Both $F_{[1,1],+}$ and $F_{[3,5],+}$ fit the $F_{[1,1],+}$ data very well for all numbers of sets of trajectories (rows 1 and 2). This indicates the classes of ODE models with different configurations of the same graph model are similar in terms of data fitting. Consequently, even with data sufficient to infer the correct graph model, it may be impossible to infer the specific ODE model.

5. The models from the flowering network cannot fit the random dataset (reflected Brownian motions with covariance 0.05) well. It turns out that the ODE models with 34 parameters have trouble following the highly variable 35 data points from the reflected Brownian motions. The low fitness level to the random dataset shows great redundancy in the parameters in terms of generating data points. It also indicates the fitting results to the synthetic ODE data are significant compared to fitting a random dataset.

## Discussion

### Generalization of CSP to related gene regulatory network models

The concept of CSP can be applied to many other models. We first explain this for continuous-state models, and then for discrete-state models.

**Continuous-state models.** A network model somewhat similar to ODE models is a fixed-point model. The study by Van den Bulcke et al. [31] uses a fixed-point model for gene regulatory networks. ODE models based on Michaelis–Menten and Hill kinetics and linear degradation terms are used to determine the expression level of a given gene as a function of the expression levels of other genes. Then a fixed point is produced. This can model equilibrium points, also known as resting points, of ODE models. The concept of constant sign property can be applied to fixed-point models as well. Van den Bulcke et al. [31] focuses on models for the network topology, which is not addressed in this paper.

Other continuous-state models have been used for gene regulatory networks. The study by Mendes et al. [32] simulates gene regulatory networks using a biochemical simulator called Gepasi [33], which models complex biochemical pathways using ODEs. For such biochemical systems, constant sign property discussed in this paper can be used to find the causal dependency among observed variables (e.g., mRNA abundances in the special case of gene regulatory networks). In order to avoid the difficult calibration of the parameters in ODEs, Ocone et al. [34] models the promoter by a binary state process and approximates the transcription–translation network with stochastic differential equations. Constant sign property can be easily generalized to such hybrid models by introducing a notion of monotonicity for the stochastic systems. It is worth mentioning that constant sign property is defined with directionality for causal relationship among the genes and not suitable for models based on mere correlation (e.g., graphical Gaussian models [35]).

**Discrete-state models.** One common type of discrete models used for gene regulatory networks are Bayesian networks (see, e.g., Friedman et al. [36]). Boolean networks, as a special case of Bayesian networks, are used to capture qualitative gene regulation (see, e.g., Liang et al. [37]), for which constant sign property can be defined based on the monotonicity of the boolean functions. The study by Husmeier [38] evaluates a dynamic Bayesian network inference algorithm using simulated data based on an ODE model whose genetic network model is taken from Zak et al. [39] and whose equations are taken from chemical kinetics (see Chapter 22 of Atkins and de Paula [40]). Similarly, the study by Smith et al. [41] also proposes a

dynamic Bayesian network algorithm, and evaluates its performance on sampled and quantized data from a dynamic Bayesian network simulator that models different regions of the brain of songbirds regulated by their behaviors. The simulated data is generated with a small step size before being sampled, and thus resembles an ODE model simulator. For the dynamic Bayesian network gene expressions are quantized to discrete values. The constant sign property can also be applied to dynamic Bayesian network models using a partial order of the conditional distributions (e.g., stochastic dominance) of target genes given the expressions of their regulators. Husmeier [38] gives an example of a graphical model that is more detailed than the gene regulatory graph in this paper. Although both the GeneNetWeaver model and the ODE models in Husmeier [38] are based on chemical kinetic equations, one difference is that the Michaelis–Menten and the Hill kinetics in GeneNetWeaver arise from considerations of a faster time scale of the binding of TF to the promoter regions (see Alon [19]). Nevertheless, both GeneNetWeaver and the ODE models for realistic simulation in Husmeier [38] fall into the general framework of ODE models in this paper and hence the constant sign property we have proposed applies to both.

## Implication of GCSP

GCSP of an ODE model generalizes the notion of a linear dynamical system by allowing the variation of the state vector (i.e., the concentrations of molecular classes) to be nonlinear in the state vector so long as the overall effect of the most influential pathways in the molecular graph keeps the same sign (i.e., activation stays activation and repression stays repression regardless of the expression of the regulator, the target gene, or any other molecular classes). Biologically, GCSP indicates homogeneity of the gene regulatory network in the sense that the qualitative properties of gene regulation are preserved after cellular differentiation and under different external conditions. Lack of GCSP indicates significant change in regulatory functions after cellular differentiation and under different external conditions. Note that GCSP is more likely to hold for the subnetwork of a small number of genes compared to a larger network.

## Limitation of infinitesimal CSP

The definitions of CSP proposed in this paper focus on short time behavior. Over short time periods, the paths with the smallest number of hops dominate. Often the shortest paths have the strongest influence, as seen in Example 2. But in some cases the shortest paths could be weaker than some slightly longer paths, and if the longer paths have an opposite sign, then the focus on short time and shortest paths can be misleading, because the longer paths will take over quickly after the brief initial dominance by the shortest paths. In the extreme case of a complete molecular graph, where every molecular class has a (possibly tiny) regulatory effect on every other molecular class, the gene regulatory graph defined in this paper would be determined by only the direct edges in the molecular graph and all the actual biological pathways would be entirely ignored. This also shows the importance of network sparsity.

## Conclusion

Gene regulatory networks are modeled at different abstraction levels with tradeoff between accuracy and tractability. Graph models with signed directed edges provide circuit-like characterization of gene regulation, while ODE models quantify detailed dynamics for various molecular classes. The constant sign property proposed in this paper connects the two types of models by identifying a set of conditions under which ODE models correspond to a single graph model, and provides a deeper understanding of the context-dependent and time-varying nature of gene regulatory networks. A class of ODE models for a given graph model based on

the source code of a popular software package GeneNetWeaver is described in detail and shown to satisfy the global constant sign property. Exploration of data fitting of one ODE model to the data generated from another shows better fit when two models have the same graph model.

## Supporting information

**S1 Appendix. Basic model of gene interaction.** A brief review on both graph models and ODE models is given here.
(PDF)

**S2 Appendix. Random model for production functions used in GeneNetWeaver.** Specific module generation and parameter ranges in GeneNetWeaver are described here.
(PDF)

**S3 Appendix. Proof of Eq (6).** A proof of the equation involving the partial derivatives of the solution of dynamical systems.
(PDF)

## Author Contributions

**Conceptualization:** Xiaohan Kang, Bruce Hajek.

**Formal analysis:** Xiaohan Kang, Bruce Hajek.

**Funding acquisition:** Bruce Hajek, Yoshie Hanzawa.

**Investigation:** Xiaohan Kang, Bruce Hajek.

**Methodology:** Xiaohan Kang, Bruce Hajek.

**Software:** Xiaohan Kang.

**Supervision:** Bruce Hajek, Yoshie Hanzawa.

**Visualization:** Xiaohan Kang.

**Writing – original draft:** Xiaohan Kang, Bruce Hajek.

**Writing – review & editing:** Xiaohan Kang, Bruce Hajek, Yoshie Hanzawa.

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
