## [Decision Letter · Decision Letter 0]

10 Mar 2020

PONE-D-20-02246

From graph topology to ODE models for gene regulatory networks

PLOS ONE

Dear Dr. Kang,

Thank you for submitting your manuscript to PLOS ONE. After careful consideration, we feel that it has merit but does not fully meet PLOS ONE’s publication criteria as it currently stands. Therefore, we invite you to submit a revised version of the manuscript that addresses the points raised during the review process.

The reviewers point out to the need to deepen the level of description and clarifying the concepts behind your analysis and results. Compare the performance of the present method with different methods, as well as highlighting the novelty and usefulness of this approach.

We would appreciate receiving your revised manuscript by Apr 24 2020 11:59PM. To enhance the reproducibility of your results, we recommend that if applicable you deposit your laboratory protocols in protocols.io, where a protocol can be assigned its own identifier (DOI) such that it can be cited independently in the future. For instructions see: http://journals.plos.org/plosone/s/submission-guidelines#loc-laboratory-protocols

We look forward to receiving your revised manuscript.

Kind regards,

Enrique Hernandez-Lemus, Ph.D.

Academic Editor

PLOS ONE

Journal Requirements:

Reviewers' comments:

Reviewer's Responses to Questions

**Comments to the Author**

1. Is the manuscript technically sound, and do the data support the conclusions?

Reviewer #1: Partly

Reviewer #2: Partly

2. Has the statistical analysis been performed appropriately and rigorously? 

Reviewer #1: No

Reviewer #2: N/A

3. Have the authors made all data underlying the findings in their manuscript fully available?

Reviewer #1: No

Reviewer #2: Yes

4. Is the manuscript presented in an intelligible fashion and written in standard English?

Reviewer #1: Yes

Reviewer #2: Yes

5. Review Comments to the Author

Reviewer #1: Overall, most given explanations are too brief and for this reason the ideas introduced are difficult to understand. The paper should be extended - at the moment it is anyway quite short.

Importantly, there is no deeper discussion of the obtained results (also a discussion section is missing). At least 3-4 of discussion should be added.

The paper is currently in a draft stage and required more work.

Introduction:

The related literature is not properly cited or discussed. Please add more background information about this. In total, about 40 papers should be cited (currently only 18 are cited).

It is unclear to me how the authors define a 'gene regulatory network'. Frequently this is done wrong, as discussed here

PMID: 25364745

Importantly, a gene networks refer to all possible types of molecular interactions, including transcriptional regulation or protein interaction. If it would only refer to the former it would be a transcriptional regulatory network.

Furthermore, the modeling perspective of such networks has been discussed in detail in

PMID: 25221572

It is unclear of GeneNetWeaver is essential for the analysis of if different tools could be used, e.g.,

PMID: 16438721

This point should be made clear.

GeneNetWeaver is an old model. Is there no newer development in this area?

Fitting results

This section is entirely unclear. I guess a regression is performed? What is the used sample size? What is R^2? What regularization has been used? How is the fit evaluated? How is dealt with overfitting?

Conclusions

The insights provided are shallow.This problem related back to my points I made above.

What does the CSP mean biologically?

The paper mentions no word that usually GRN do have many more than 5 genes? Can one extend the model to 100 genes?

Reviewer #2: The paper under review is unfocused and rather weak.

The main purpose seems to be to define carefully an ODE model called GeneNet Weaver that has been used to to produce data used in DREAM challenges. Then the main result seems to be observation in three lines that the ODE's generated by this model have monotone dependence on their arguments and therefore are

consistent with a network graphical model with signed edges.

However, there is insufficient justification why this class of models reflects biological reality. For instance, the module 1 uses product between all activators and deactivators which assumes that all activators must be bound for transcription to happen. This is not true in many promoter regions where either activator can activate the gene. If these modules can be then considered independently, this must be explained and stated.

In line 205 a statement about a sum being considered 1 vs zero is puzzling, and left hanging without any explanation. Yet this seems to be. a crucial point of the discussion about relationship between modules 2 and 3.

There is no comparison with older similar models of gene regulation based on statistical mechanics and occupancy or promoters. In particular, line of papers

Ackers GK, Johnson AD, Shea MA., Quantitative model for gene regulation by lambda phage repressor., Proc Natl Acad Sci U S A. 1982 Feb;79(4):1129-33

Shea MA, Ackers GK., The OR control system of bacteriophage lambda. A physical-chemical model for gene regulation., Mol Biol. 1985 Jan 20;181(2):211-30.

should be cited and discussed. In particular,

Gedeon, Mischaikow K, Patterson K, Traldi E.,

When activators repress and repressors activate: a qualitative analysis of the Shea-Ackers model.

Bull Math Biol. 2008 Aug;70(6):1660-83. doi: 10.1007/s11538-008-9313-6. Epub 2008 Jul 22.

has shown that the Shea-Ackers type model does not have to have the monotonicity property that is claimed in this paper for GeneNet Weaver models.

Discussion on what assumptions on GeneNet Weaver models guarantee the monotonicity must be made.

It is not clear what the case study communicates. Artificial data is generated and then fit with ODE models. This has no bearing on biology.

I recommend either a rejection or a major revision for this paper.

6. PLOS authors have the option to publish the peer review history of their article (what does this mean?). If published, this will include your full peer review and any attached files.

Reviewer #1: No

Reviewer #2: No

---

## [Decision Letter · Decision Letter 1]

9 Jun 2020

From graph topology to ODE models for gene regulatory networks

PONE-D-20-02246R1

Dear Dr. Kang,

We’re pleased to inform you that your manuscript has been judged scientifically suitable for publication and will be formally accepted for publication once it meets all outstanding technical requirements.

Kind regards,

Enrique Hernandez-Lemus, Ph.D.

Academic Editor

PLOS ONE

Additional Editor Comments (optional):

Reviewers' comments:

Reviewer's Responses to Questions

**Comments to the Author**

1. If the authors have adequately addressed your comments raised in a previous round of review and you feel that this manuscript is now acceptable for publication, you may indicate that here to bypass the “Comments to the Author” section, enter your conflict of interest statement in the “Confidential to Editor” section, and submit your "Accept" recommendation.

Reviewer #2: All comments have been addressed

2. Is the manuscript technically sound, and do the data support the conclusions?

Reviewer #2: Yes

3. Has the statistical analysis been performed appropriately and rigorously? 

Reviewer #2: I Don't Know

4. Have the authors made all data underlying the findings in their manuscript fully available?

Reviewer #2: Yes

5. Is the manuscript presented in an intelligible fashion and written in standard English?

Reviewer #2: Yes

6. Review Comments to the Author

Reviewer #2: (No Response)

7. PLOS authors have the option to publish the peer review history of their article (what does this mean?). If published, this will include your full peer review and any attached files.

Reviewer #2: No

---

## [Editor Report · Acceptance letter]

17 Jun 2020

PONE-D-20-02246R1 

From graph topology to ODE models for gene regulatory networks 

Dear Dr. Kang:

I'm pleased to inform you that your manuscript has been deemed suitable for publication in PLOS ONE. Congratulations! Your manuscript is now with our production department. 

Kind regards, 

on behalf of

Prof. Enrique Hernandez-Lemus 

Academic Editor

PLOS ONE